# The Metabolic Footprint of Systemic Effects in the Blood Caused by Radiotherapy and Inflammatory Conditions: A Systematic Review

**DOI:** 10.3390/metabo13091000

**Published:** 2023-09-09

**Authors:** Karol Jelonek, Katarzyna Mrowiec, Dorota Gabryś, Piotr Widłak

**Affiliations:** 1Center for Translational Research and Molecular Biology of Cancer, Maria Sklodowska-Curie National Research Institute of Oncology, Gliwice Branch, 44-100 Gliwice, Poland; katarzyna.mrowiec@gliwice.nio.gov.pl; 2Department of Radiotherapy, Maria Sklodowska-Curie National Research Institute of Oncology, Gliwice Branch, 44-100 Gliwice, Poland; dorota.gabrys@gliwice.nio.gov.pl; 32nd Department of Radiology, Medical University of Gdańsk, 80-210 Gdańsk, Poland; piotr.widlak@gumed.edu.pl

**Keywords:** ionizing radiation, head and neck cancer, radiotherapy, inflammation, metabolomics, lipidomics

## Abstract

Response to radiotherapy (RT) includes tissue toxicity, which may involve inflammatory reactions. We aimed to compare changes in metabolic patterns induced at the systemic level by radiation and inflammation itself. Patients treated with RT due to head and neck cancer and patients with inflammation-related diseases located in the corresponding anatomical regions were selected. PubMed and Web of Science databases were searched from 1 January 2000 to 10 August 2023. Twenty-five relevant studies where serum/plasma metabolic profiles were analyzed using different metabolomics approaches were identified. The studies showed different metabolic patterns of acute and chronic inflammatory diseases, yet changes in metabolites linked to the urea cycle and metabolism of arginine and proline were common features of both conditions. Although the reviewed reports showed only a few specific metabolites common for early RT response and inflammatory diseases, partly due to differences in metabolomics approaches, several common metabolic pathways linked to metabolites affected by radiation and inflammation were revealed. They included pathways involved in energy metabolism (e.g., metabolism of ketone bodies, mitochondrial electron transport chain, Warburg effect, citric acid cycle, urea cycle) and metabolism of certain amino acids (Arg, Pro, Gly, Ser, Met, Ala, Glu) and lipids (glycerolipids, branched-chain fatty acids). However, metabolites common for RT and inflammation-related diseases could show opposite patterns of changes. This could be exemplified by the lysophosphatidylcholine to phosphatidylcholine ratio (LPC/PC) that increased during chronic inflammation and decreased during the early phase of response to RT. One should be aware of dynamic metabolic changes during different phases of response to radiation, which involve increased levels of LPC in later phases. Hence, metabolomics studies that would address molecular features of both types of biological responses using comparable analytical and clinical approaches are needed to unravel the complexities of these phenomena, ultimately contributing to a deeper understanding of their impact on biological systems.

## 1. Introduction

Radiotherapy (RT) uses ionizing radiation (IR) to damage DNA and other components of cells, which ultimately leads to the destruction of their ability to reproduce [1]. IR is mainly directed at tumor volume, but it also damages neighboring healthy tissue, which may result in radiation toxicity. Radiation exposure activates several biological mechanisms. Radiation-induced damage causes the death of target cells and changes in intercellular signaling [2], inflammation [3], immune responses [4], tissue repair processes [5], and the growth [6] of residual cells to compensate for cell loss [7]. The patient’s response to RT is assessed based on observable clinical responses. Clinical symptoms associated with radiation-induced damage to normal tissues are usually reflected as early and late radiotoxic reactions [8]. Early radiation reactions are syndromes of functional and morphologic disturbances in the cells and intercellular spaces of tissues during and within 3 months after radiotherapy [9]. Early radiation toxicity is mostly pronounced in tissues with rapid cellular regeneration, which is associated with a reduction of functional cells that are removed as part of the normal tissue cycle and not replaced by damaged stem cells. Such unrepaired damage causes cell death during the first few cell divisions after the damage. Besides cell death, IR induces various cell signaling pathways. These include activation of inflammatory cytokines and inflammatory cytokine cascades [10], vascular damage [11], and coagulation cascade signaling [12]. The patient’s systemic response to radiation treatment was already measured at different ‘omics’ levels in body fluids [13,14,15,16,17,18,19]. Among these, metabolomics has great clinical potential as it directly reflects phenotypic and functional changes and correlates with the changes observed at the level of transcripts and proteins [20]. Blood metabolome changes during RT with different time-related patterns. In early response, the number of metabolites with significantly changed levels is high. Over time some of these changes are diminished, while others become chronic. 

RT is an effective treatment for patients suffering from head and neck cancer (HNC). Moreover, this cancer is among a few malignancies where the independent RT alone remains a modality of radical treatment. Therefore, this malignancy represents a valid clinical model for molecular radiobiology [21]. Aggressive treatment of HNC (e.g., dose-escalated hypofractionation or accelerated fractionation) is usually required in advanced cancer cases, which might be associated with an increased risk of treatment toxicity. The RT dose escalation trials in HNC treatment [22,23,24,25,26] relied on functional imaging modalities that assisted with target delineation and allowed researchers to push this boundary further. On the other hand, more recently, there is also an observable trend of dose [27,28,29,30,31] and volume [32,33] de-escalation to reduce the subsequent toxicity. Another reported reason for increased toxicities in RT patients is due to the unexpected RT errors where patients receive a higher dose or/and volume of radiation [34]. The major symptom of early radiation toxicity in HNC patients is mucositis, which is usually associated with inflammation and early-phase response [35], but also dermatitis [36]. The body’s metabolism could be affected not only by radiation toxicity within the irradiation field but also by weight loss during and after treatment, which is related to difficulties with swallowing, pain, and appetite loss [37,38,39,40]. More than 50% of HNC patients lose more than 5% of their body weight at the time of initial treatment [41]. Importantly, weight loss alone can induce changes in lipid and amino acid metabolism, energy metabolism, and dietary composition, which was documented for healthy persons [42].

Inflammation is a normal response to injury or infection. Acute inflammatory response involves the release of interleukins [43] and lipid mediators [44] that can cause organ dysfunction. One of the well-known lipid mediators is arachidonic acid, which, during inflammation, is converted to prostaglandins [45]. Acute inflammation usually heals quickly, but chronic inflammation persists and harms healthy tissue. It occurs when the immune system is constantly activated and attracts more leukocytes to the inflamed tissues. Chronic inflammation can affect different body systems and cause various diseases [46]. Inflammatory diseases, both acute and chronic, also influence blood metabolic patterns, which raises the question of the similarity of metabolic changes induced by RT and those associated with inflammatory conditions. One could hypothesize that because inflammation is an important component of radiation toxicity developed during RT, metabolic features observed at the systemic level in the blood of patients undergoing RT and patients suffering from inflammation-related diseases may show similar patterns. 

RT-related changes in blood metabolome profiles are best studied in the case of HNC compared to other types of cancers and non-cancer diseases. Therefore, in this systematic review, we aimed to compare the blood metabolome changes observed after irradiation in HNC patients to inflammation-related changes caused by diseases located in the corresponding anatomical regions (i.e., head and neck area or skin). Diseases associated with acute and chronic inflammation were selected to enable the comparison of metabolic features of radiation response with both types of inflammatory conditions. We wanted to know whether early radiation-induced metabolic changes follow patterns characteristic of acute or chronic inflammatory diseases. 

## 2. Materials and Methods

This systematic review was elaborated following the preferred reporting items for systematic reviews and meta-analyses (PRISMA) statements [47]. 

### 2.1. Search Strategy

We systematically retrieved PubMed and Web of Science databases for articles published between 1 January 2000 and 10 August 2023. The search terms are presented in Table 1. Records were filtered to exclude case reports, meta-analyses, letters, editorials, meeting abstracts, authors’ replies, books, reviews, and articles nonrelevant according to inclusion criteria. Non-English articles were also removed. Two investigators [KJ and KM] conducted an independent preliminary screening based on the titles and abstracts. Full texts were retrieved and prepared to assess eligibility in the next step. Disagreements were resolved by involving the third researcher [PW] in the event of inconsistencies.

### 2.2. Eligibility Criteria

The inclusion criteria for selecting eligible studies were as follows: (a) studies provided a measure of association between control samples and samples from patients with inflammatory diseases; (b) studies provided a measure of association between patients before and after radiotherapy; and (c) studies were peer-reviewed publication. On the contrary, studies were excluded if they: (a) reported inflammation diseases in combination with other diseases; (b) involved drug supplementation or other treatments; (c) additionally prefiltered serum or plasma and used only fraction; (d) involved children (age < 18); (e) did not include statistical analysis to compare the groups. Two researchers (KJ and KM) independently identified the eligible studies, and their discrepancies were resolved by involving the third researcher (PW). 

### 2.3. Data Extraction and Preparation

A standardized data extraction form was developed to record data from each eligible study in connection with the aims of this systematic review. The following study characteristics were recorded from each eligible study: name of the first author, publication year, inflammatory or RT model, number of patients and controls, sample type, method of analysis, and analyzed metabolites. For each analyzed metabolite, the status of change (significant increase, significant decrease, or no significant change) was recorded. The level of significance was chosen by the authors of the particular study based on their data.

### 2.4. Pathway Analysis

Metabolic pathways were associated with metabolites with a net significant vector of changes different than zero in all screened studies. Over-represented metabolic pathways were associated with metabolites differentiating between chronic or acute inflammatory disease and healthy control as well as between pre-RT and post-RT statuses using the enrichment analysis on the MetaboAnalyst platform (https://www.metaboanalyst.ca/MetaboAnalyst/ModuleView.xhtml (accessed on 14 July 2023)). Obtained enriched pathways and their connections were further analyzed in Cytoscape. The DyNet addon was used to compare two networks and find interacting nodes [48].

## 3. Results

### 3.1. Study Selection

Figure 1 presents the study selection flow chart. The applied search strategy retrieved 1151 articles in the Pubmed and Web of Science databases. Duplicate screening excluded 530 articles, while abstract screening excluded another 497 articles. The full texts of the remaining 124 articles were reviewed concerning the study selection criteria, which resulted in 25 studies included in this systematic review.

### 3.2. Study Characteristics

The characteristics of the 25 eligible studies are summarized in Table 2. Specifically, 4 studies were related to acute inflammatory diseases (AID), 14 studies were related to chronic inflammatory diseases (CID), and 7 studies were related to radiotherapy (RT). 

About 1000 compounds present in at least one study were identified; however, only 7 metabolites were commonly analyzed in each of three groups of studies. The numbers of metabolites measured in the studied groups are presented in Figure 2.

Most of the metabolites in the selected studies showed no statistically significant changes (according to the methodology applied by the authors), or changes reported in one study were “counterbalanced” by data from another report. Therefore, to extract a metabolite affected by the analyzed condition, we calculated a net significant change for each metabolite. A net significant change was equal to zero if none of the studies showed a statistically significant change or the number of reports with statistically significant increases and decreases was equal. Then, metabolites for which a net significant change was different from zero were used in further analyses. Information about specific compounds analyzed in each of the 25 eligible studies is presented in Appendix A.

### 3.3. Systemic Effects Caused by Acute Inflammatory Diseases

We found four studies with acute inflammatory diseases (summarized in Table 2). In all cases, the analyzed disease was periodontitis. One of the studies utilized GC-MS [50], while the rest used LC-MS/MS approach to analyze serum metabolome. There was no single metabolite that was measured in all four publications. Two studies analyzed arginine and its methylated derivatives: one showed no significantly affected metabolites [52], while the other showed two significantly affected compounds (out of six measured) [51]. Another study performed by LC-MS/MS revealed 28 metabolites (out of 51 measured) that were significantly increased in periodontitis [49]. These metabolites were related to omega-3 and omega-6 polyunsaturated fatty acids (PUFAs) and PUFA-metabolites of linoleic acid, arachidonic acid (AA), eicosapentaenoic acid (EPA) and docosahexaenoic acid (DHA). Another study performed by GC-MS revealed six significantly affected metabolites [50]. In general, out of 62 metabolites measured in studies related to acute inflammatory diseases, 36 compounds had a net significant change different than zero. Among metabolites with reduced levels in the sera of patients were SDMA, 2,5-dihydroxybenzaldehyde, 2-deoxyguanosine, adipic acid, and glutathione. On the other hand, among metabolites with increased levels in sera of patients were allo-inositol, ADMA, two eicosanoids, five HDHA, five HEPE, hepoxilin A3, five HETE, two HODE, lipoxin A4, oxylipin 13, prostaglandin D2, protectin D1, PUFAs (arachidonic acid and its 20-COOH derivative, docosahexaenoic acid, and eicosapentaenoic acid), and urea.

### 3.4. Systemic Effects Caused by Chronic Inflammatory Diseases

Search for chronic inflammation-related diseases in the region of the head and neck showed disorders related only to the skin. Since the structure of the skin is similar in all regions of the human body, we included in our study skin diseases located in different areas, which revealed 14 studies summarized in Table 2. Most of the studies were focused on psoriasis (10 investigations), two involved atopic dermatitis, and two single studies targeted hidradenitis suppurativa and dermatomyositis. Most of the examinations applied LC-MS/MS (11); two of them used GC-MS, and one study used NMR. The majority of studies analyzed broad metabolic profiles (10 studies), some of them specific lipid profiles (3 studies), and one of them analyzed bioactive lipid mediators [58]. Out of 882 compounds measured in studies related to chronic inflammatory diseases, 153 metabolites showed a net significant change different than zero. For the majority of these compounds, the net significant change was equal to −1 or 1 (e.g., one paper showed a significant change in a specific compound, either increase or decrease, while other(s) papers showed no significant change of the same compound). Six metabolites showed a net significant change equal to −2 (i.e., two more studies with particular metabolites showing significantly lower levels in disease in comparison to the number of studies presenting no change or upregulation): arachidonic acid, 12-HETE, lysoPE (20:4), lactosylceramide (d18:1–12:0), and two PCs (36:4 and 38:5), while six metabolites showed the net significant change equal to +2: four LPCs (16:0, 18:1, 18:0 and 22:6), palmitic amide, and oleamide. 

### 3.5. Comparison of Systemic Effects Caused by Acute and Chronic Inflammatory Diseases

The number of studies available in the literature regarding metabolomic patterns in either type of inflammatory disease (compared to healthy controls) and the number of addressed metabolites is substantially different (Figure 3A), and only six compounds (out of 183 affected in either group) were affected in both acute and chronic inflammatory diseases (Figure 3B). Among 31 metabolites measured in both types of studies, there were 24 compounds affected in either condition (a net change different than 0): 23 metabolites in acute conditions and 7 metabolites in chronic conditions (Figure 3C). Six metabolites (arachidonic acid, docosahexaenoic acid, and four hydroxyeicosatetraenoic acids) were affected in both inflammatory conditions. However, these compounds showed increased levels in the serum/plasma of patients with acute inflammation while decreased levels in the serum/plasma of patients with chronic inflammation. This indicated rather limited similarity of metabolic pattern characteristics for acute and chronic inflammatory conditions.

The comparison of specific compounds may not reflect the essence of the difference between acute and chronic inflammation due to markedly different sets of metabolites addressed in both types of studies. Therefore, to enable a more systemic look at metabolic changes, metabolites were annotated with their corresponding pathways that were compared between conditions. The pathway analysis included metabolites with net significance different than zero (as shown in Figure 3B). Over-represented metabolic pathways associated with metabolites differentiating between chronic inflammatory disease or acute inflammatory disease and relevant healthy controls are presented in Figure 4. Among over-represented pathways associated with metabolites affected in both types of inflammatory conditions were arachidonic acid metabolism (the most affected in both groups), linolenic acid metabolism, and urea cycle (Figure 4 A,B). The Cytoscape joint pathway analysis showed the network of 39 pathways associated with metabolites affected by inflammatory conditions (Figure 4C). Among them, only two pathways (Arginine and proline Metabolism and Urea Cycle) were linked to both types of inflammatory conditions, which indicated different patterns of metabolic changes associated with both types of inflammatory conditions.

### 3.6. Systemic Effects Caused by Radiotherapy

Seven studies related to systemic metabolomic effects caused by RT were retrieved, all of them concerned patients irradiated due to HNC (summarized in Table 2). One study analyzed a global profile of serum metabolites using the NMR approach [71], two studies addressed the global metabolite profile of serum using GC-MS [68,69], and one study addressed plasma amino acids by GC-MS [70]. Choline-containing phospholipids were addressed in two studies based on MS, either LC-MS/MS [19] or MALDI-MS [67]. Another study analyzed a panel of 180 lipids and small metabolites using a targeted MS-based approach [18]. In general, all these studies addressed 358 compounds in total, of which 76 compounds were significantly affected (a net change different than zero) when pre-RT and post/within-RT samples were compared. Additionally, in four studies, samples from irradiated patients were collected more than one time post/within RT, which allowed us to study the kinetics of serum metabolome response to ionizing radiation [18,19,67,71]. This is especially important since the metabolomic changes are dynamic in time (most of the early changes might be compensated later and some even beyond the original pre-RT levels), and not all studies analyzed samples collected at the same time points after the start of RT. In the study that analyzed choline phospholipids, there was a massive decrease in LPC and PC levels two weeks after the start of RT and their subsequent increase one month after the end of RT [67]. The other three studies analyzed samples collected during RT and right after RT. They also revealed differences in metabolic patterns between both time points yet did not show a direct compensation effect, which suggested the requirement of longer post-RT times for the compensation [18,19,71]. 

### 3.7. Comparison of Systemic Effects Caused by Radiotherapy and Inflammation-Related Diseases

The number of studies available in the literature regarding metabolomic patterns in irradiated humans and patients with either type of inflammatory disease, as well as the number of addressed metabolites, is substantially different (Figure 5A). Among them, there were 21 metabolites affected both after RT and any inflammatory conditions (Figure 5B).

A set of 214 compounds was measured in both types of studies (Figure 5A). Among them, 53 metabolites were affected by RT, and 44 metabolites were affected by the inflammation-related disease (Figure 5C). A subset of 21 metabolites affected in both conditions included glutamine, dimethylglycine, sebacic acid, 9 phosphatidylcholines, 5 lysophosphatidylcholines, 2 sphingomyelins, oleamide, and urea. Importantly, the only metabolite common for radiation response and acute inflammation was urea, whose level increased in AID and decreased in RT. Other metabolites common for RT and inflammation were characteristic of chronic inflammation. It is noteworthy that except for oleamide (increased in both conditions), levels of common metabolites showed an opposite direction of changes in response to radiation and upon chronic inflammation. Importantly, after RT, levels of LPCs decreased, and levels of PCs decreased, while during inflammation, levels of LPCs increased, and levels of five (out of nine) PCs increased. However, although serum concentrations of LPCs decreased in the first stage of RT, their levels increased in the second stage of RT (healing stage or late effects of RT) [67], which represented a pattern similar to that observed during chronic inflammation. Moreover, glutamine and sebacic acid increased in RT and decreased inflammation, while dimethylglycine decreased in RT and increased in inflammation.

In order to enable a more systemic look at metabolic changes linked to radiation and inflammation, metabolites were annotated with their corresponding metabolic pathways (metabolites with net significant changes different than zero were included). Over-represented metabolic pathways associated with metabolites differentiating between pre-RT and post-RT samples as well as between inflammatory disease and healthy controls, are presented in Figure 6. Among the top 25 overrepresented pathways associated with metabolites affected by inflammation (Figure 6A) and radiation response (Figure 6B), there were eight pathways common for both types of conditions (with the urea cycle as the top ranking). Moreover, when the network of pathways associated with radiation or inflammation-affected metabolites was revealed, there were 16 common pathways, which included the metabolism of ketone bodies and butyrate, metabolism of glycerolipids, metabolism of certain amino acids (Arg, Pro, Gly, Ser, Met, Ala, Glu), metabolism of betaine, urea cycle, citric acid cycle, Warburg effect, glucose-alanine cycle, mitochondrial electron transport chain, oxidation of phytanic acid and branched chain fatty acids (Figure 6C). Two of these sixteen are common for RT and ID pathways (arginine and proline metabolism and urea cycle) and were linked to both types of inflammatory conditions; the rest were specific only to chronic conditions. Although a limited number of common metabolites was identified, several metabolic pathways associated with response to radiation and inflammatory conditions were identified using a pathway analysis approach. However, one should be aware that specific metabolites associated with these common pathways could show the opposite direction of changes in either type of condition, which was described in the previous paragraph. Nonetheless, patterns of RT-related changes strongly correlate with the time of sample collection, which could correspond either to the early phase of response or the late/healing phase of response [18,19,67,71], which further complicates the comparison to inflammatory conditions.

## 4. Discussion

Inflammation is a critical constituent of normal responses to injury or infection [72]. During acute inflammatory response, large amounts of interleukin and lipid mediators are released and play an important role in the pathogenesis of organ dysfunction. Arachidonic acid is released from phospholipids located in the cell membrane during the activation of an inflammatory response, and it is converted to prostaglandins [45]. Activation of this process was also observed in one of the reviewed studies within the acute inflammatory disease group, where both arachidonic acid and prostaglandin D2 were shown to have significantly changed levels in the disease [49]. Most of the acute inflammations repair relatively quickly, and metabolic homeostasis is rapidly restored. However, a wide range of inflammatory diseases exist due to chronic activation of the immune system, resulting in chronic persistent inflammation [73]. In chronic conditions, inflammation becomes more of a problem than a solution for infection, injury, or illness. Chronic inflammatory tissues continue to signal to attract leukocytes from the bloodstream. As leukocytes move from the blood to the tissues, they amplify the inflammatory response. This chronic inflammatory response damages healthy tissue in a misdirected attempt to repair and heal. Chronic inflammation can take various forms and have the potential to affect diverse body systems [74,75,76,77,78,79,80,81,82]. Here, we reviewed 14 studies related to chronic inflammation of the skin. Among serum components commonly linked to chronic inflammation were lysophospatidylcholines (LPCs) which increased in disease conditions (in contrast to phosphatidylcholines, PCs). Hence, increased LPC/PC ratio was a characteristic mark of chronic inflammation.

Systemic effects caused by acute and chronic inflammatory diseases differ at every “OME” level, including metabolome [83]. This is also visible from the current review, where only six common metabolites were affected in both types of inflammatory conditions, and all of them changed their levels in different directions in the blood of patients with acute and chronic inflammation. Therefore, there is no general metabolomic pattern of inflammation that could be compared with metabolomic patterns related to radiation response. However, the comparison of metabolomic changes induced by RT and either chronic or acute inflammation revealed several common features. The only significantly changed metabolite common for radiation response and acute inflammation was urea, whose level increased in one study related to aggressive periodontitis [50] and decreased in one study related to RT [69]. However, in this particular report, post-RT samples were collected one month after the end of RT and putatively represented the healing stage of the process, where the levels of radiation-influenced metabolites are compensated even beyond the original pre-RT levels. The affected LPC/PC ratio was the most characteristic difference when common features of chronic inflammation and radiation response were addressed. In chronic inflammation, an increased LPC/PC ratio was observed, while a decreased LPC/PC ratio was linked with response to RT of HNC patients. This observation is contrary to previous observations in whole-body irradiated mice, where an increased ratio of LPCs to PCs was proposed as a radiation biomarker that generally indicates a pro-inflammatory response and phospholipase A activity [84]. Furthermore, the analysis of patients irradiated due to prostate cancer revealed that changes in the LPC/PC ratio depend on the irradiation scheme and dose: the LPC/PC ratio increased after conventional intensity-modulated RT and decreased after hypofractionated accelerated RT [85]. It is also important to note that serum concentrations of LPCs that increased in the initial phase of response to RT were elevated in the later post-RT samples (late RT effects or the healing stage) [67], which resembled the pattern of inflammation more closely.

Large differences in analytical approaches applied to asses serum/plasma metabolic profiles in all retrieved studies are the major limitation of the current analysis. Therefore, only two studies that used the same quantitative targeted MS-based approach to analyze serum metabolite profiles in patients with atopic dermatitis [63] and HNC patients treated with RT [18] were retrieved. Another important limitation of the study is the fact that the radiation response was reported in cancer patients since no metabolomic study exists for patients exposed to RT due to noncancer conditions. Hence, the characteristics of control for radiation response (i.e., pre-RT samples of cancer patients) and inflammatory conditions (i.e., samples of healthy humans) are markedly different. For example, patterns of metabolites linked to energy metabolism (e.g., glycolysis, glycogenesis, Warburg effect, TCA cycle, pyruvate metabolism, and mitochondrial electron transfer chain) were significantly different in pre-RT samples of HNC patients and samples of healthy individuals [69]. Therefore, only differences between addressed conditions (i.e., radiation exposure or inflammation-related disease) and “specific controls” were taken into account. One should also be aware that RT, in addition to toxic reactions induced in the irradiated tissues, may affect the patient’s metabolomic balance in many other ways (e.g., weight loss frequently observed during and after RT). Nonetheless, new metabolomics studies using a comparable clinical and analytical model to compare directly the effects of irradiation and inflammation would be of great importance to elucidate the impact of both conditions on biological systems.

## 5. Conclusions

This review underscores the delicate balance of inflammation, the intricate repercussions of ionizing radiation exposure, and the potential overlaps and disparities in their metabolic signatures. Inflammation presents its dual nature as a vital acute response promoting tissue repair and metabolic equilibrium and as a detrimental chronic process that damages healthy tissues due to sustained immune activation. A critical observation emerges from the comparison of metabolomic responses to ionizing radiation and acute/chronic inflammation. The network of pathways revealed common pathways associated with radiation and inflammation-affected metabolites, which included the urea cycle, the metabolism of certain amino acids (Arg, Pro, Gly, Ser, Met, Ala, Glu), metabolism of ketone bodies and butyrate, metabolism of glycerolipids, Warburg effect, citric acid cycle, and mitochondrial electron transport chain. While some common features exist, the most distinguishing difference lies in the LPC/PC ratio, which fluctuates inversely in response to radiation and chronic inflammation. However, this pattern is not uniform across different radiation schemes and doses, underscoring the complexity of the response. In light of these findings, there is an urgent need for a deeper exploration of metabolomic patterns and their underlying mechanisms in both inflammation and radiation responses, especially the need for comprehensive studies directly comparing these effects, utilizing comparable analytical approaches and controls. This would shed further light on the intricate interplay between inflammation, radiation, and their metabolic implications.

## Figures and Tables

**Figure 1 metabolites-13-01000-f001:**
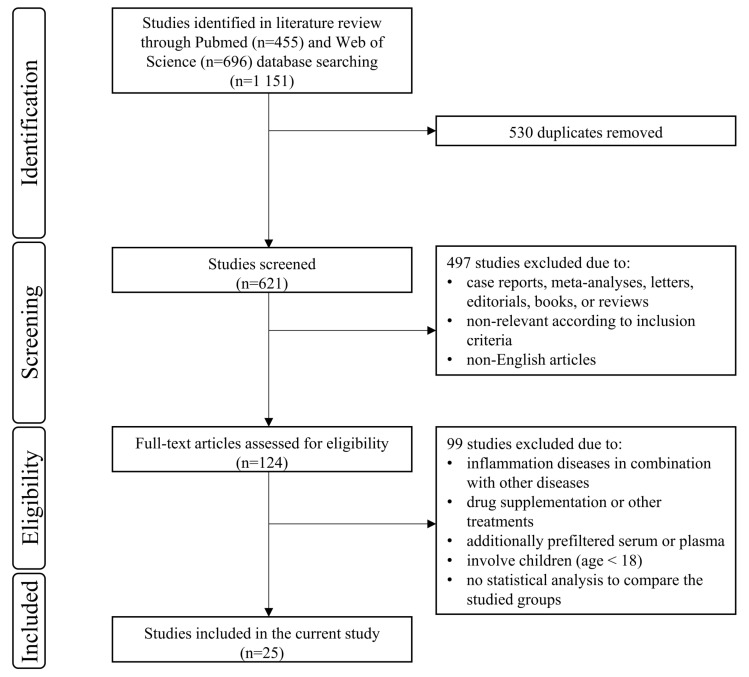
Flowchart of study selection. It presents the process by which relevant studies were retrieved from the databases, assessed, selected, or excluded. Preferred reporting items for systematic reviews and meta-analyses (PRISMA) diagram was used for the study search.

**Figure 2 metabolites-13-01000-f002:**
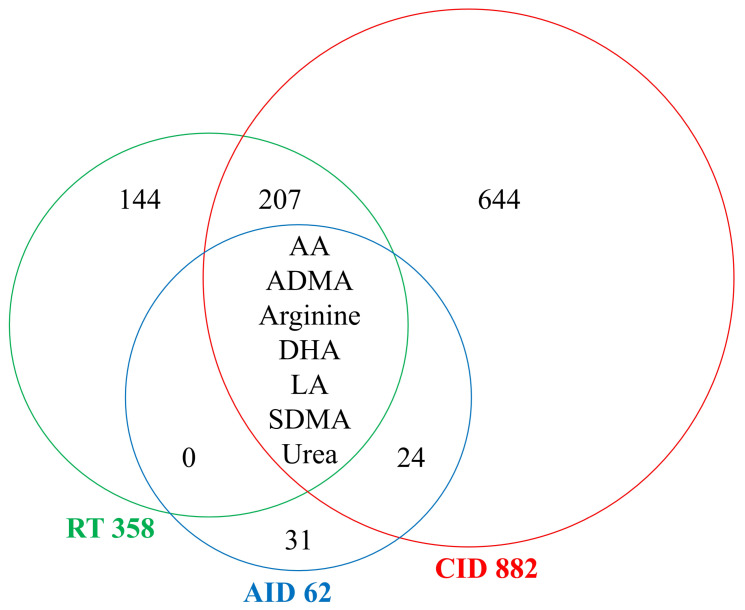
The number of metabolites identified in analyzed studies. AA—arachidonic acid; ADMA—asymmetric dimethylarginine; AID—acute inflammatory diseases; CID—chronic inflammatory diseases; DHA—docosahexaenoic acid; LA—linoleic acid; RT—Radiotherapy; SDMA—symmetric dimethylarginine.

**Figure 3 metabolites-13-01000-f003:**
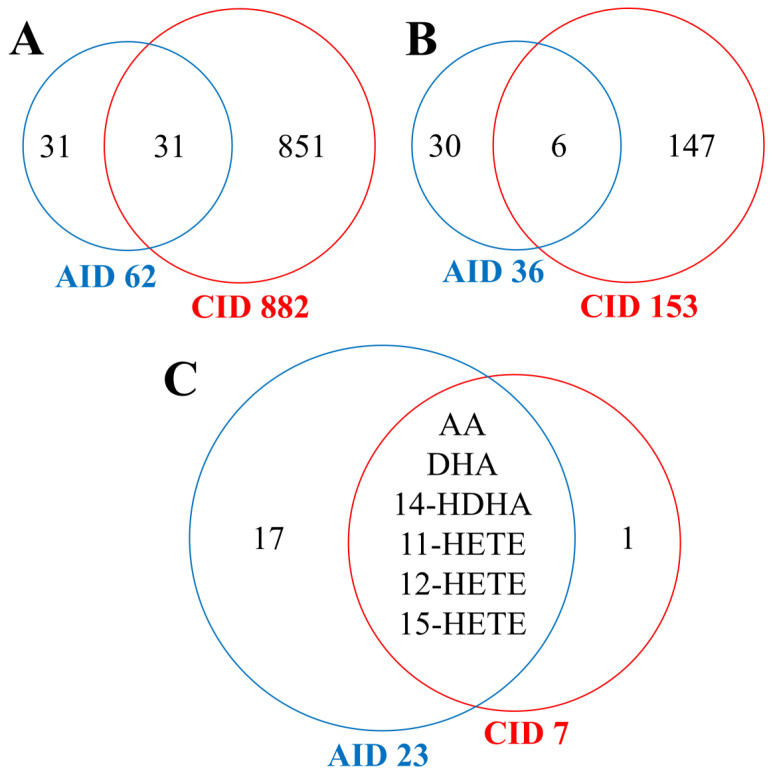
Numbers of metabolites analyzed in studies related to acute and chronic inflammatory disease. (**A**) Numbers of all measured compounds. (**B**) Numbers of all metabolites which significantly changed levels were noted between disease conditions and relevant controls. (**C**) Numbers of metabolites discriminating inflammatory conditions and controls in the subset of compounds measured in both types of inflammation. AA—arachidonic acid; AID—acute inflammatory diseases; CID—chronic inflammatory diseases; DHA—docosahexaenoic acid; HDHA—hydroxy docosahexaenoic acid; HETE—hydroxyeicosatetraenoic acid.

**Figure 4 metabolites-13-01000-f004:**
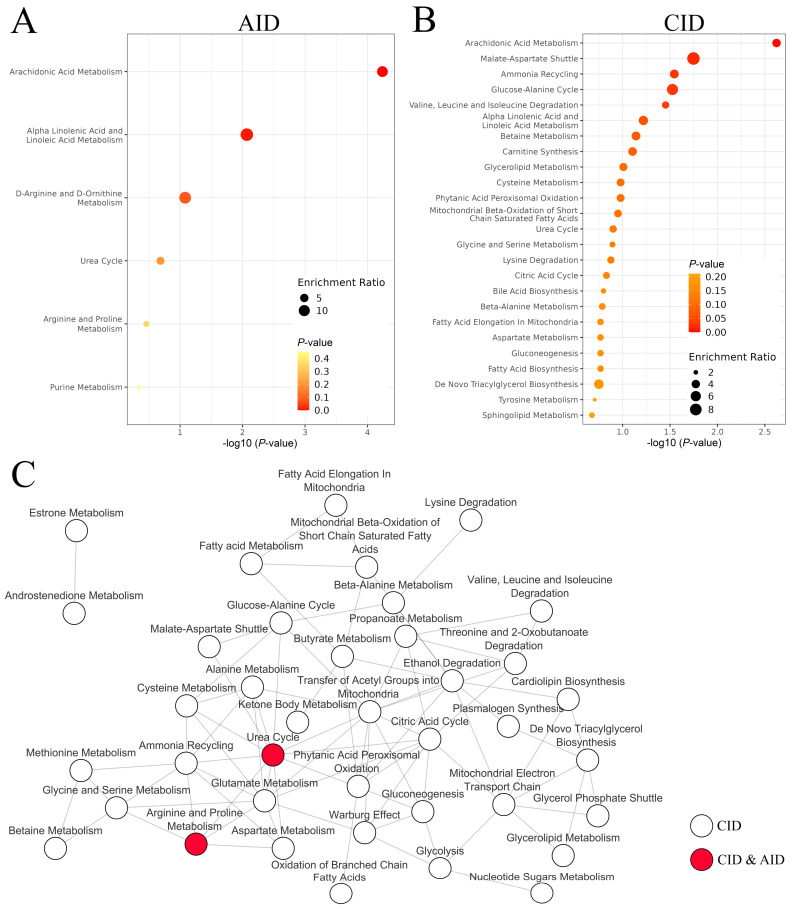
Pathways associated with metabolites affected by inflammatory diseases. Overview of top metabolite sets enriched in acute inflammation (**A**) and chronic inflammation (**B**) according to the MataboAnalyst analysis. The higher the dot in Enrichment Ratio, the more metabolites were present in a given pathway. (**C**) Network of pathways associated with inflammation-related metabolites (according to the Cytoscape analysis). AID—acute inflammatory diseases; CID—chronic inflammatory diseases.

**Figure 5 metabolites-13-01000-f005:**
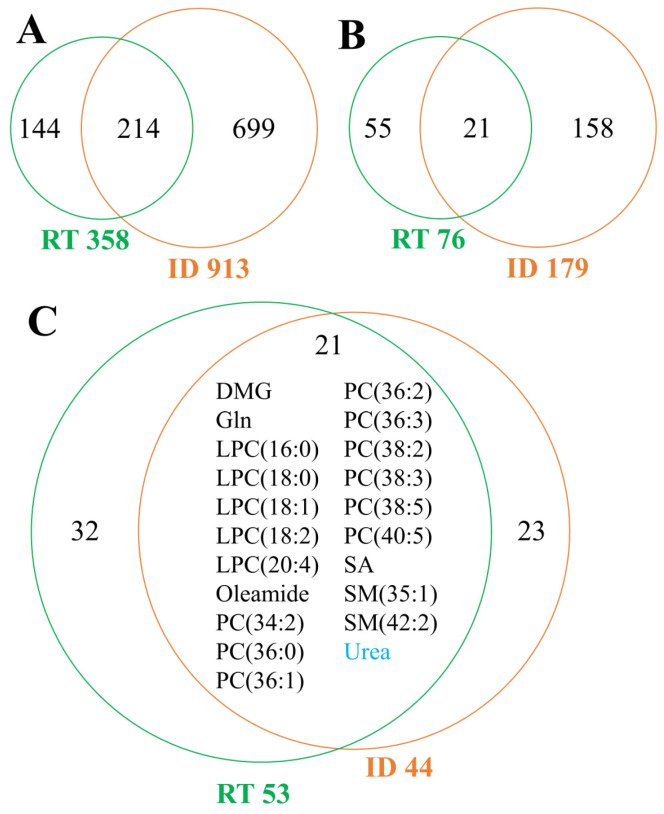
Numbers of metabolites analyzed in studies related to RT and inflammatory diseases. (**A**) Numbers of all measured compounds in either type of study (both types of inflammation-related diseases treated collectively). (**B**) Numbers of all metabolites that significantly changed levels were noted after irradiation (RT) or in inflammation-related disease (ID). (**C**) Numbers of metabolites affected by RT and discriminating inflammatory conditions from healthy controls in the subset of compounds measured in both types of studies. Urea marked with blue was affected by acute inflammation, and other compounds were affected by chronic inflammation. DMG—dimethylglycine; Gln—glutamine; ID—inflammatory diseases; LPC—lysophosphatidylcholine; PC—phosphatidylcholine; RT- radiotherapy; SA—sebacic acid; SM—sphingomyelin.

**Figure 6 metabolites-13-01000-f006:**
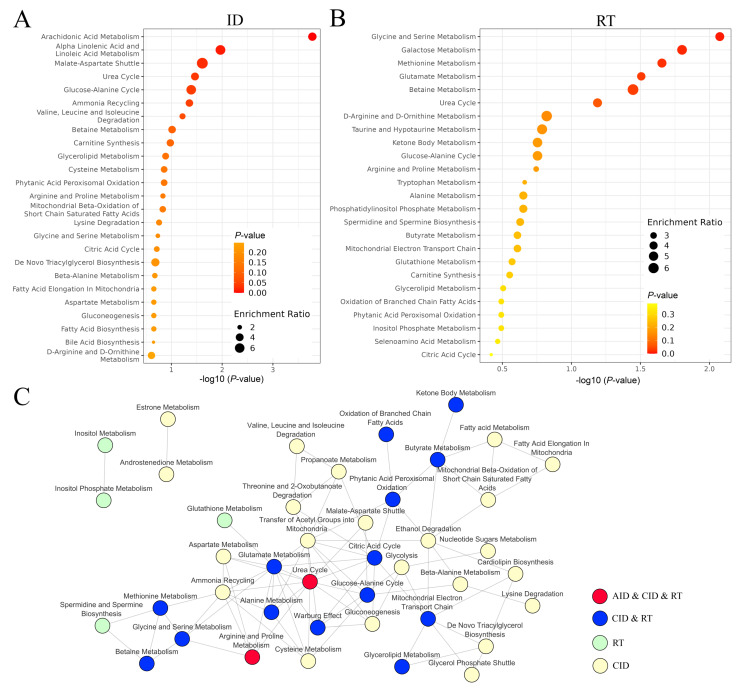
Pathways associated with metabolites affected by radiation and inflammatory diseases. Overview of top metabolite sets enriched in inflammatory conditions (**A**) and after irradiation (**B**) according to the MataboAnalyst analysis. The higher the dot in Enrichment Ratio, the more metabolites were present in a given pathway. (**C**) Network of pathways associated with irradiation and inflammation-related metabolites (according to the Cytoscape analysis). ID—inflammatory diseases; AID—acute inflammatory diseases; CID—chronic inflammatory diseases; RT—radiotherapy.

**Table 1 metabolites-13-01000-t001:** The search terms were used to retrieve the studies from the databases. The successful record had to consist of at least one term from each column.

	AND				
**OR**	- metabolite	- radiotherapy	- serum	- human	- mass spectrometry
	- metabolites	- radiation	- sera	- humans	- MS
	- metabolome	- dermatitis	- plasma	- patient	- gas chromatography
	- metabonome	- mucositis	- plasmas	- patients	- GC
	- metabolic	- laryngitis	- blood		- liquid chromatography
	- metabolomic	- odynophagia			- LC
	- metabolomics	- esophagitis			- capillary electrophoresis
	- lipid	- pharyngitis			- CE
	- lipids - lipidome - lipidomics	- tonsillitis - aphtha - otitis			- matrix-assisted laser desorption ionization-time of flight
	- lipidomic	- epiglottitis - sinusitis - pulpitis			- matrix-assisted laser desorption/ionization-time of flight
		- periodontitis			- MALDI-TOF
		- gingivitis			- triple quadrupole
		- xerostomia			- QQQ
		- shingles			- ion trap
		- erysipelas			- IT
		- sialadenitis			- quadrupole ion trap
		- angina			- QIT
					- orbitrap
					- Fourier transform ion cyclotron resonance
					- FTICR
					- nuclear magnetic resonance
					- NMR
					- magnetic resonance spectroscopy
					- MRS

**Table 2 metabolites-13-01000-t002:** Studies presenting the influence of inflammatory diseases or radiotherapy on human blood metabolome.

Clinical Model	Controls ^1^ (n)	Cases ^2^ (n)	Type of Material	Method of Analysis	Analyzed Metabolites	Reference
Acute Inflammatory Disease						
AgP	19	19	serum	LC-MS/MS	n3 and n6 PUFAs	[49]
AgP	20	20	serum	GC-MS	global profile	[50]
periodontitis	24	21	serum	LC-MS/MS	arginine and derivatives	[51]
periodontitis	20	20	serum	LC-MS/MS	arginine and derivatives	[52]
Chronic Inflammatory Disease						
psoriasis	10	10	serum	GC-MS	global profile	[53]
psoriasis	75	75	serum	LC-MS	global profile	[54]
psoriasis	32	85	serum	GC-MS	fatty acid profile	[55]
psoriasis	45	45	plasma	LC-MS/MS	global profile	[56]
psoriasis	7	7	serum and plasma	LC-MS/MS	BLMs	[57]
psoriasis	30	60	serum and plasma	LC-MS/MS; GC-MS	BLMs	[58]
psoriasis	12	12	plasma	LC-MS	global profile	[59]
psoriasis	45	50	serum	LC-MS	global profile	[60]
psoriasis	28	28	serum	LC-MS	lipid profile	[61]
psoriasis	7	7	serum	NMR	global profile	[62]
AD	24	25	serum	LC-MS/MS	global profile ^3^	[63]
PM or DM	12	13	serum	LC-MS/MS; GC-FID	global profile	[64]
HS	73	60	plasma	LC-MS/MS	global profile	[65]
AD	47	30	serum	LC-MS/MS	Cer, Sph, LPC, PC, SM, and LPE	[66]
Radiotherapy ^4^						
CAIR (1.8; 72)	66	66	serum	MALDI-MS	global profile	[67]
CAIR (1.8; 72)	20	20	serum	GC-MS	global profile	[68]
CAIR; CFRT; (1.8–2.5; 62.5–72)	18	18	serum	LC-MS/MS	global profile ^3^	[18]
CAIR (1.8; 72)	10	10	serum	GC-MS	global profile	[69]
CFRT (2; 70)	28	28	plasma	GC-MS	amino acids	[70]
CAIR; CFRT; MAN (1.8–3; 51–72)	106	106	serum	NMR	global profile	[71]
CAIR; CFRT; MAN (1.8–3; 51–72)	45	45	serum	LC-MS/MS	global profile	[19]

^1^ healthy controls in case of inflammatory disease or pre-RT samples in case of RT; ^2^ inflammatory disease or within/post-RT samples in case of RT; ^3^ Biocrates p180 targeted assay; ^4^ In the “Clinical model” column specified are Intensity-Modulated Radiation Therapy (IMRT) schemes with dose per fraction (Gy) and total doses (Gy) in the parenthesis. AD—atopic dermatitis; AgP—aggressive periodontitis; BLMs—Bioactive lipid mediators; CAIR—continuous accelerated irradiation; CFRT—conventional fractionated radiotherapy; DM—dermatomyositis; HNC—head and neck cancer; HS—hidradenitis suppurativa; MAN—Manchester scheme (accelerated hypofractionated irradiation); PM—polymyositis; PUFAs—polyunsaturated fatty acids.

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
