# Peer review of "The Metabolic Footprint of Systemic Effects in the Blood Caused by Radiotherapy and Inflammatory Conditions: A Systematic Review"

_metabolites, 2023, doi:10.3390/metabo13091000_

Round 1
Reviewer 1 Report
Author in the manuscript (Systematic Review) analyzed the impact of radiation therapy (RT) uses ionizing radiation (IR) and inflammation on metabolites. Radiotherapy (RT) is commonly used to treat cancer, especially solid tumors. Radiation not only kills or slows the growth of cancer cells, it can also affect nearby healthy cells. Damage to healthy cells can cause side effects and metabolic changes.
Major comments:
1. With RP patients often experience metabolic changes including weight gain, patient’s slow metabolism was observed which contribute to weight gain. The Glycine and serine metabolism, galactose and glutamate metabolism are directly influence with weight gain. The control study is not included in the study i.e., impact of radiation is almost like the amount of nutrients lost during cooking, freezing, canning and other food safety methods. The impact of the systematic review is not influential or does not impact the treatment in future. Instead oncometabolites and response to radiotherapy is impactful. In fact, the therapeutic potential of RT alone and in multimodal combinations with surgery, chemotherapy, and targeted drug therapy has increased considerably during the past decades.
2. RT dose escalation is also an important issue that need to address, which is missing in the review article.
3. The most common errors reported include patients receiving the wrong dose of radiation, the wrong site being treated also need to address in the review.
4. Therefor lots of factors are involved in radiation therapy and metabolic changes.
The quality of language is good, simple and readable
Reviewer 2 Report
hello
thank you very much for an interesting paper;
introduction -ok
aim of review -ok
material and method section -ok
results -good
good diagrams and flow chart - very well described
systematic effects and RTH affects well described
excellent comparison criteria
good discussion
In my opinion the following review should be accepted in its current form
Author Response
Dear Reviewer#2,
thank you for your time and your acceptation of the review in the current form.
Reviewer 3 Report
This is an interesting study on the analysis of metabolic footprints after radiotherapy. The importance of the study is related to the identification of clinical biomarkers of response to RT.
I have several comments regarding this manuscript:
1. The Abstract should be reorganised:
1.1 please specify particular metabolic pathways involved in RT response and revealed in this study;
1.2 The Authors said that "the studies showed different metabolic patterns of acute and chronic inflammatory diseases" - It is not entirely clear from the abstract, whether head and neck cancer refers to chronic or acute conditions?
1.3 The aim of the study should be written more clearly.
1.4. In general Abstract should immediately give readers a clear understanding of this study.
2. It may be worth adding “blood” to the title of the manuscript (not obligatory)
3. Head and neck cancer includes a number of heterogeneous forms of tumors, why did the authors choose HNC?
4. The Authors mentioned that “RT-related changes in blood metabolite profiles are best studied in the case of this cancer”. If so, and the field of knowledge is well-known, what is the relevance of the current research?
5. p.2, raw76 – please change to interleukins
6. Why the study period is 1/1/2000-2/10/2023? Is there any relevant studies after February 2023? I think the Authors should extend the search period for articles from 1/1/2000-2/10/2023 to May or July.
7. The conclusions section can be improved - specify the specific metabolomic pathways revealed in this research
8. Only 45 References in the manuscript and looks like 9 of them are self-cited studies. This is 20% of all REFs. The editors should manage if it is ok and does not violate the requirements.
Round 2
Reviewer 3 Report
The Authors have addressed my comments.